# Estimation and Validation of Sub-Pixel Needleleaf Cover Fraction in the Boreal Forest of Alaska to Aid Fire Management

**Anushree Badola** [1,*] 🄳, **Santosh K. Panda** [1,2] 🄳, **David R. Thompson** [3] 🄳, **Dar A. Roberts** [4] 🄳,
**Christine F. Waigl** [5] 🄳 and **Uma S. Bhatt** [1] 🄳

1    Geophysical Institute, University of Alaska, Fairbanks, AK 99775, USA; skpanda@alaska.edu (S.K.P.);
     usbhatt@alaska.edu (U.S.B.)
2    Institute of Agriculture, Natural Resources and Extension, University of Alaska, Fairbanks, AK 99775, USA
3    Jet Propulsion Laboratory, California Institute of Technology, Pasadena, CA 91109, USA;
     david.r.thompson@jpl.nasa.gov
4    Department of Geography, University of California, Santa Barbara, CA 93106, USA; daroberts@ucsb.edu
5    International Arctic Research Center, University of Alaska, Fairbanks, AK 99775, USA; cwaigl@alaska.edu
*    Correspondence: abadola@alaska.edu

**Abstract:** Wildfires, which are a natural part of the boreal ecosystem in Alaska, have recently increased in frequency and size. Environmental conditions (high temperature, low precipitation, and frequent lightning events) are becoming favorable for severe fire events. Fire releases greenhouse gasses such as carbon dioxide into the environment, creating a positive feedback loop for warming. Needleleaf species are the dominant vegetation in boreal Alaska and are highly flammable. They burn much faster due to the presence of resin, and their low-lying canopy structure facilitates the spread of fire from the ground to the canopy. Knowing the needleleaf vegetation distribution is crucial for better forest and wildfire management practices. Our study focuses on needleleaf fraction mapping using a well-documented spectral unmixing approach: multiple endmember spectral mixture analysis (MESMA). We used an AVIRIS-NG image (5 m), upscaled it to 10 m and 30 m spatial resolutions, and applied MESMA to all three images to assess the impact of spatial resolution on sub-pixel needleleaf fraction estimates. We tested a novel method to validate the fraction maps using field data and a high-resolution classified hyperspectral image. Our validation method produced needleleaf cover fraction estimates with accuracies of 73%, 79%, and 78% for 5 m, 10 m, and 30 m image data, respectively. To determine whether these accuracies varied significantly across different spatial scales, we used the McNemar statistical test and found no significant differences between the accuracies. The findings of this study enhance the toolset available to fire managers to manage wildfire and for understanding changes in forest demography in the boreal region of Alaska across the high-to-moderate resolution scale.

**Keywords:** spectral unmixing; needleleaf; hyperspectral; MESMA

## 1. Introduction

Boreal forests in Alaska are found between the Brooks Range in the north and the Alaska Range in the south, covering an area of 43 million hectares [1,2]. Evergreen coniferous vegetation which mostly include black spruce (*Picea mariana*) and white spruce (*Picea glauca*) dominate boreal forests, particularly in interior Alaska which is the heartland of wildfires. Climate is the primary driver for wildfires in boreal Alaska [3]. Between 1976 and 2016, the annual temperature of Alaska increased by 0.3 degrees Celsius per decade and is expected to rise by 1–2 degrees Celsius by 2050 [4–6]. According to Bieniek et al. (2020), lightning has increased by 17% throughout Alaska over the last 30 years. These changes are influencing wildfire events and increasing the fire frequency, severity and burn acreage [7]. In the last two decades (2003–2022: 12.9 million hectares), wildfires in Alaska have burned around twice as many hectares than the previous two

decades (1983–2002: 6.7 million hectares) [8,9]. Though wildfires are a natural part of the boreal ecosystem, recycling soil nutrients and renewing forest health, increasing fire frequency, severity and burned acreage have far-reaching environmental and societal impacts. Some of the adverse impacts include loss of habitat and subsistence resources, risk to communities at the wildland–urban interface, high cost of fire-fighting and restoration, and disproportionate carbon emissions [5,10]. To a large extent, fire spread and intensity are dependent on the vegetation or fuel types. Needleleaf vegetation/fuel are more flammable and spread fire more efficiently compared to broadleaf vegetation/fuel due to their resin content and low-lying canopy structure that serves as a ladder fuel leading to severe crown fires [11]. The maps of the needleleaf vegetation distribution are important at all three stages of fire management. Prior to the fire season, they can help land managers identify high fire risk areas to employ fuel management practices such as building fire breaks, tree thinning, removing dead fuels, etc. During an active fire, these maps can serve as input for fire spread modeling and forecasting near real-time fire spread and behavior. Post-fire, they can help understand the impacts of fire on the ecosystem, forest recovery and demography.

Remote sensing is a proven approach to map vegetation types or classes. Pixel-level mapping is very popular in the remote sensing community, where each pixel is mapped as a vegetation class [8,12–16]. In reality, a pixel can contain more than one class; in that case, the total pixel reflectance will be the combination of the reflectance from all classes present within the pixel. In a boreal landscape, a pixel can contain both needleleaf and broadleaf vegetation. However, a pixel-level vegetation product will map the pixel as one class that dominates in the pixel. For sub-pixel vegetation mapping, i.e., the estimation of the different vegetation fraction in a pixel, one can employ spectral unmixing, also known as spectral mixture analysis (SMA), an approach to map a vegetation fraction where the algorithm calculates the proportion of each class within a pixel. SMA considers the spectrum of a single pixel as a weighted sum of the constituent spectra of classes or endmembers [17], providing sub-pixel level estimates of the vegetation class fraction. Multiple endmember spectral mixture analysis (MESMA) [18] is an advanced SMA method that assumes that an image is composed of large numbers of different endmembers or classes, but a pixel can be composed of a subset of endmembers. Hence, MESMA allows a large number of endmembers across the scene, but each pixel is modeled independently with a different number and type of endmembers. MESMA was used to map green vegetation, non-photosynthetic vegetation (NPV), and soil in Santa Monica Mountains California, USA, using AVIRIS data [18]. Fernández-García et al. (2021) applied MESMA to Landsat data to study habitat diversity over Cantabrian Mountains located in the northwest of the Iberian Peninsula [19]. Fernandez-Manso et al. (2016) used MESMA to map the burn severity using Landsat images over Sierra del Teleno in Northern Spain using green vegetation (GV), non-photosynthetic vegetation and ash (NPVA), and soil as endmembers [20]. In boreal Alaska, it can be challenging to distinguish a needleleaf pixel from a mixed pixel due to the lower spectral separability between the classes. This challenge provides an opportunity to test the MESMA algorithm for needleleaf mapping in boreal Alaska. Furthermore, there are several existing global hyperspectral space missions which provide data at a coarser spatial resolution (30 m or 60 m). Examples of these missions include the Earth Surface Mineral Dust Source Investigation (EMIT) [21] which has 285 spectral bands (381–2493 nm) and 60 m spatial resolution; the Environmental Mapping and Analysis Program (EnMap) [22], which has 228 spectral bands (420–2450 nm) and 30 m spatial resolution; and the PRecursore IperSpettrale of the application mission (PRISMA) [23], which has 220 spectral bands (400–2500 nm) and 30 m spatial resolution. Additionally, a new mission called Surface Biology and Geology (SBG) [24] is planned to have 217 spectral bands (80–2500 nm) and 30 m spatial resolution. These missions will increase the availability of the hyperspectral data for a variety of applications, including detailed vegetation mapping at the regional scale. These global sensors have a coarser spatial resolution compared to aerial hyperspectral images such as AVIRIS-NG; therefore, it is also important to evaluate the performance

of MESMA in mapping major vegetation classes at different spatial resolutions [25]. This study focuses on sub-pixel level needleleaf vegetation mapping at different spatial scales using AVIRIS-NG data, addressing the following research questions:

A. Does MESMA have the potential to estimate the needleleaf fraction in a mixed boreal vegetation with reasonable accuracy?

B. Does the spatial resolution of a hyperspectral image influence the estimation of needleleaf fraction?

C. How can we validate spectral unmixing estimations at different spatial scales?

## 2. Materials and Methods

The methodology comprises two main components: pixel unmixing and validation, as illustrated in Figure 1. We collected endmembers (needleleaf, broadleaf, and NPV) from hyperspectral imagery (AVIRIS-NG). Then, we performed spectral unmixing and validated the results using methods, elaborated in the following sections.

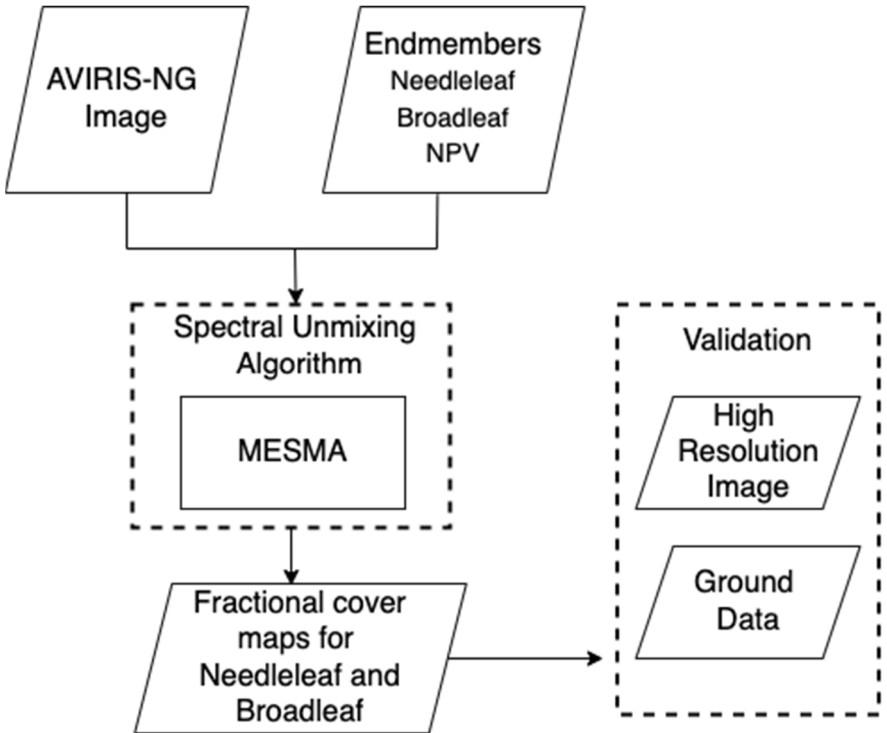

**Figure 1.** Processing workflow for needleleaf and broadleaf fraction mapping and validation.

### 2.1. Study Area

The National Science Foundation (NSF) established the Long-Term Ecological Research Program (LTER) Network in 1980 to conduct ecological studies and collect long-term datasets to analyze environmental change. Bonanza Creek Experimental Forest (BCEF) is one of the LTER sites in Alaska. It is located in interior Alaska (64.70°N, −148.30°W), approximately 30 km southwest of Fairbanks, covering an area of 5053 ha. For this study, we selected a test site within BCEF (Figure 2), where the AVIRIS-NG scene was available. This region lies between the Brooks Range in the north and the Alaska Range in the south, which blocks the coastal air masses; hence, the area experiences cold winters as well as warm and dry summers. The study area experiences short growing seasons (100 days or less), and the air temperature ranges from −50 °C in January to over +33 °C in July, with a long-term average annual temperature of −3 °C. The mean annual precipitation is approximately 269 mm, 30% of which is in the form of snowfall [1]. The study area includes both upland and lowland regions with a variety of vegetation types [26]. The soils are immature, ranging from cold soils with shallow permafrost in the lowlands to

warm and well-drained soils in the uplands. Lowlands and north-facing slopes are covered by moss-dominated black spruce, while aspen, birch, and white spruce mainly grow in uplands and south-facing slopes.

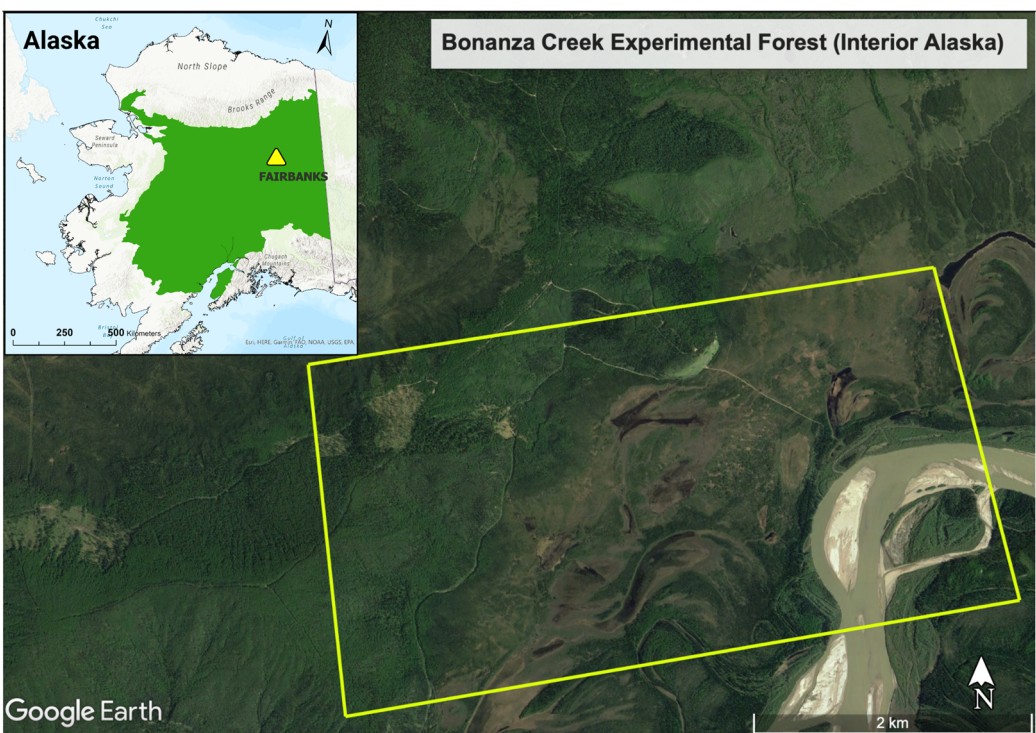

**Figure 2.** Study area: Bonanza Creek Experimental Forest (BCEF), located 30 km southwest of Fairbanks in interior Alaska. Boreal forest area represented in green on inset map. A yellow boundary delineates the study area within BCEF.

### 2.2. Field Data Collection

This study required extensive fieldwork to validate the fraction product generated in this study. It involved conducting on-site surveys to record the proportion and distribution of various species within the study area. Specifically, we collected the percentage cover and count of tree species within a plot. We surveyed 40 plots of 10 m × 10 m size during 2019 (Figure 3a,b) and two plots of ~1000 m² in size in 2022 (Figure 3c) using a Trimble Real-Time Kinematic (RTK) Global Positioning System (GPS) unit that offers millimeter-level positional accuracy. In the field, we recorded the relative proportion of key vegetation species in 40 (10 m × 10 m) plots. We subdivided the 2022 field plots into smaller plots and recorded the broadleaf and needleleaf tree counts. We also noted the tree species that dominate the top canopy. We did not see any evidence of forest damage or anthropogenic change such as timber harvesting since 2018 (image year), ensuring that the use of image data and field data collected during different times is reasonable (Tables 1 and 2).

**Table 1.** List of image datasets used in this study.

| Data | Scene Identifier | Acquisition Date | Spatial Resolution | Bands |
|---|---|---|---|---|
| AVIRIS-NG | ang20180723t200207 | 23 July 2018 | 5 m | 425 |
| SkySat | 20190629_002107_ssc10_u0002 | 29 June 2019 | 0.5 m | 4 |
| HySpex | 20210803_BC | 3 August 2021 | 1 m | 459 |

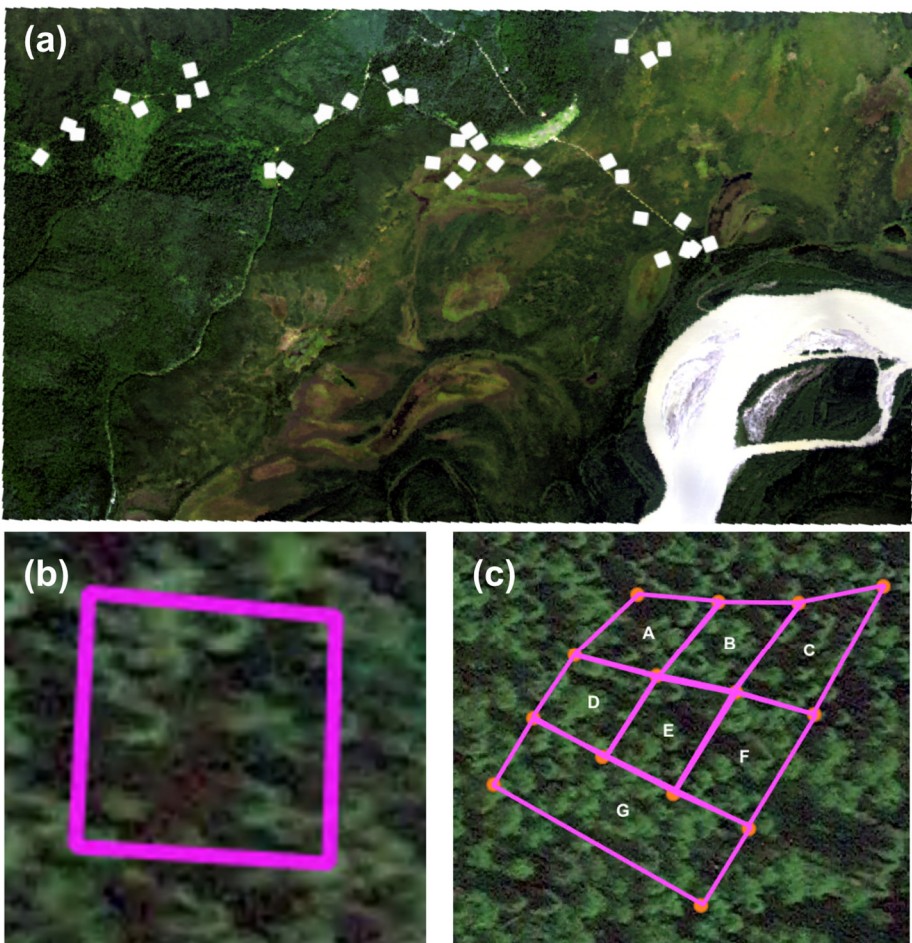

**Figure 3.** Ground data collected for validation: (**a**) white polygons show the location of 40 (10 m × 10 m) plots; (**b**) a single (10 m × 10 m) plot; (**c**) a ~1000 m² plot subdivided into 7 subplots (A–G). The fairly dense forest setting made it difficult to set up a 30 m × 30 m square plot in the field.

**Table 2.** List of field data used in this study.

| Plot Data | Instrument | Data Collection Time | Data Collected |
|---|---|---|---|
| 10 m × 10 m (40 plots) | Trimble RTK-GPS | Summer 2020 and 2021 | Vegetation composition, canopy cover, diameter and height |
| ~1000 m² (2 plots) | Trimble RTK-GPS and Garmin Handheld GPS device | Summer 2022 | Needleleaf tree count |

### 2.3. Data Preprocessing

We used the atmospherically and radiometrically corrected level-2 AVIRIS-NG data (Table 1) acquired as a part of the Arctic-Boreal Vulnerability Experiment (ABoVE) [27]. The data have 425 bands, 5 m pixel resolution, and were collected on 23 July 2018. We removed bands with excessive noise due to poor radiometric calibration and bands dominated by water vapor and methane absorption [28]. Table 3 lists the removed bands. For this study, we used a subset of the AVIRIS-NG flight line (Table 1). All the preprocessing was performed in ENVI classic 5.3 [29].

**Table 3.** Bands removed from AVIRIS-NG hyperspectral data.

| Bands | Wavelength (nm) | Remarks |
|---|---|---|
| 196–210 | 1353.55–1423.67 | Water vapor absorption bands |
| 288–317 | 1814.35–1959.60 | Water vapor absorption bands |
| 408–425 | 2415.39–2500.00 | Noise due to poor radiometric calibration and strong water vapor and methane absorption |

### 2.4. Endmember Selection

Selecting appropriate endmembers is a critical step in spectral unmixing. In our study area, the primary vegetation classes consist of spruce (needleleaf) and birch (broadleaf), which serve as two crucial endmembers [8,15]. Additionally, we considered including other endmembers in our analysis, such as shrubs (including broadleaf and evergreen shrubs), non-photosynthetic vegetation (NPV), and soil. However, we found that the spectral contrast between broadleaf trees and shrubs was low, which made it difficult for the algorithm to distinguish between them. Our primary goal was to map needleleaf vegetation, which are highly flammable, so we combined the broadleaf trees and shrubs into a single class.

When we ran the spectral unmixing algorithm with the combined broadleaf/shrub class, needleleaf, NPV, and soil as endmembers, we found that the algorithm could not accurately distinguish the soil and NPV due to the low spectral contrast between them and could only map one of them. As NPV includes dead branches, leaf litter, and dry vegetation, which are highly flammable and important for wildfire management, we decided to drop soil from consideration and ultimately selected needleleaf, broadleaf, and NPV as our endmembers. Selecting these endmembers allowed us to focus on mapping the highly flammable fuels (needleleaf vegetation and NPV) that most impact fire spread. Spectral processing was performed in the Visualization and Image Processing for Environmental Research (VIPER) Tools 2 (beta) software as an extension of the ENVI software [30]. Figure 4 shows the endmember spectra selected for this study.

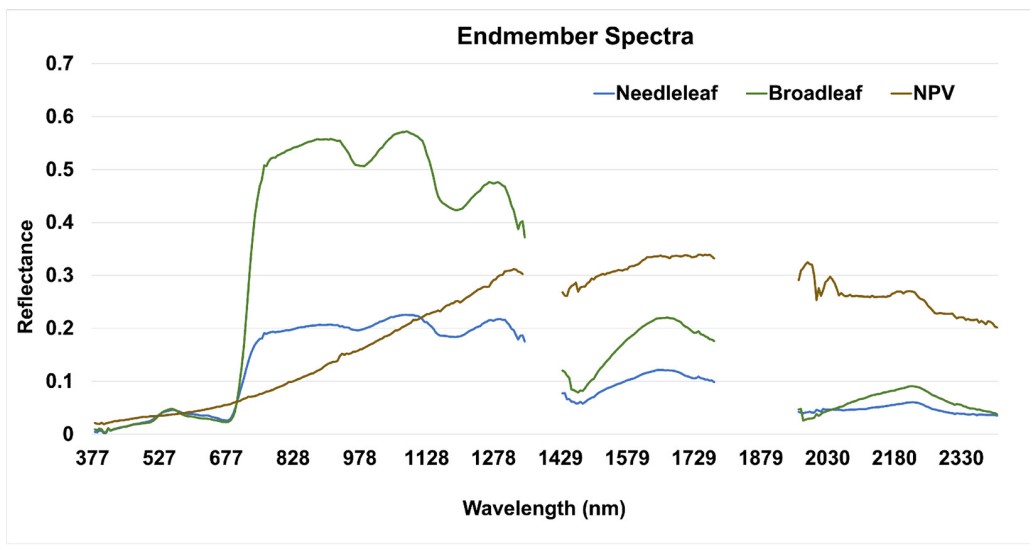

**Figure 4.** Endmember spectra of needleleaf, broadleaf and NPV. The graph shows the average spectra for all three endmembers used in the spectral unmixing model.

### 2.5. Spectral Unmixing

This study focused on mapping the highly flammable fuels (needleleaf vegetation) at sub-pixel level. We assumed that every pixel in the AVIRIS-NG image could be modeled by a linear combination of two types of vegetation (broadleaf and needleleaf) and NPV (Figure 5). We applied multiple endmember spectral mixture analysis

(MESMA) [31] by analyzing all of the potential endmember combinations for each pixel, starting with one endmember model: Needleleaf, Broadleaf, and NPV, and two endmember models: Needleleaf-Broadleaf, Needleleaf-NPV, and Broadleaf-NPV. Upon using the three-endmember model, we found an increase in the number of unclassified pixels to 25%, which indicated that the model was not effectively capturing the spectral variability in the data. As a result, we decided to use only one and two endmember models since the majority of the pixels (over 90%) were modeled by them. We constrained the minimum and maximum permitted endmember fractional values between 0.00 and 1.00, and a maximum allowable RMSE of 0.025.

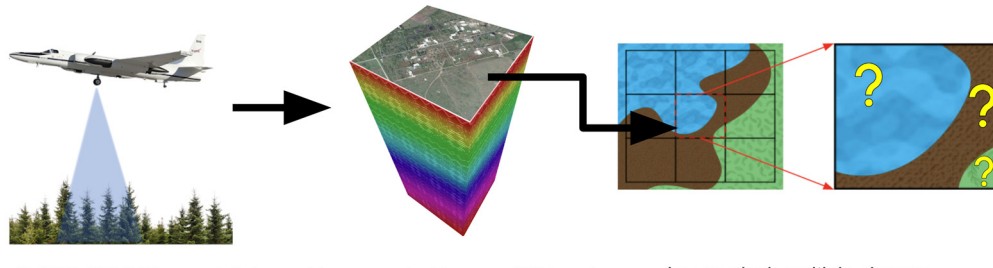

AVIRIS-NG (airborne data)     Hyperspectral image: 425 bands     Image pixel: multiple classes

**Figure 5.** A pixel can contain more than one class at different proportions marked in yellow. Spectral unmixing approach is used to estimate the proportions of classes within a pixel.

In the MESMA algorithm, the selection of endmembers is critical for the accurate spectral unmixing of mixed pixels. However, in some cases, there may be low spectral contrast between the different endmember types, which can result in the algorithm picking one endmember and adding shade to compensate for brightness. To address this issue and improve the accuracy of spectral unmixing, a shade constraint is often applied in MESMA [32]. The shade constraint limits the maximum shade fraction in the spectral unmixing process, guiding the algorithm to select a bright endmember for bright stands and a dark endmember for dark stands. By doing so, it can effectively differentiate between different vegetation types and minimizes the misclassification caused by low spectral contrast [33]. In this study, we set the maximum allowable shade fraction to 0.30, which means that the algorithm was optimized to select the needleleaf for darker vegetation and broadleaf for brighter vegetation.

We evaluated the performance of all endmember models using the RMSE and recorded the model with the lowest RMSE for each pixel. The output contained an RMSE band, a band with the models, and fraction bands containing broadleaf, needleleaf, and NPV fraction estimates.

Shade normalization [34] is a post-processing step in spectral unmixing algorithms to remove the effects of shadows caused by topography or other features in an image. The process involves normalizing the estimated fractions of each endmember by dividing them by the sum of fractions of all endmembers in a given pixel excluding shade. This normalization ensures that the total estimated fraction of all non-shade endmembers in a pixel equals 1. Shade normalization is particularly useful in cases where the contrast between endmembers is low, and the spectral unmixing algorithm tends to select one endmember over the others [31]. We performed shade normalization on the spectral unmixed output to improve the estimated fractions of broadleaf, needleleaf and NPV in each pixel. We used VIPER Tools, Version 2 (beta) software [30] as an ENVI plugin to run MESMA and Shade normalization. We resampled the 5 m AVIRIS-NG image to 10 m and 30 m pixel sizes using cubic resampling in the GDAL warp function [35], and re-ran MESMA keeping the same parameters. We used cubic resampling as it determines the pixel value through a weighted average of the 16 closest pixels, resulting in a more accurate representation of the original data [36].

*2.6. Accuracy Assessment*

We used three methods to validate the fraction product: (1) visually using high-resolution SkySat image; (2) using 40 (10 m × 10 m) field plots; and (3) using 1 m resolution classified map derived from aerial HySpex image.

2.6.1. Visual Assessment Using High-Resolution Multispectral Data

We visually compared the fraction output images with the high-resolution SkySat data (50 cm pixel resolution) provided by Planet Labs under NASA Commercial Smallsat Data Acquisition Program [37]. We analyzed different areas of interest based on the kind of vegetation class and compared them from the field photos as well.

2.6.2. Assessments Using 10 m × 10 m Field Plots

We used the percentage cover information available from the 40 field plots (Figure 3) collected during fieldwork for assessing the performance of sub-pixel output from MESMA. In the MESMA output, we summed up the proportion of each pixel falling within the field plot boundary (ground data) and evaluated the needleleaf fraction. For this, we vectorized the pixels in the fractional output image using the "Raster pixels to polygons" tool available in QGIS [38]. We then clipped field plots from the vectorized/polygon shapefile and obtained the final clipped shapefile, as shown in Figure 6b. We compared the needleleaf fraction from MESMA output and ground data and estimated RMSE. We used the Point Sampling Tool [39] in QGIS to extract the raster values (proportions) from the output fraction images using the clipped shapefile (Figure 6b).

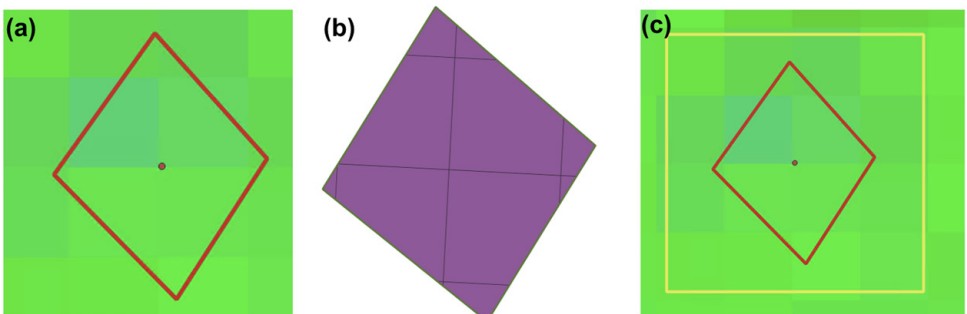

**Figure 6.** Estimating the proportion of each pixel within a (**a**) 10 m × 10 m field plot boundary (red polygon) overlaid on the fraction output raster point denotes the centroid of the plot; (**b**) pixels clipped (black boundary) that fall within the plot; and (**c**) a buffer of 10 m (yellow rectangle) created to include the variability of the vegetation around the plot.

We also considered the variability due to the positional inaccuracies in the field plot and the AVIRIS-NG image by taking a 10 m buffer from the plot centroid (Figure 6c). We summed up all the pixels whose center was located within the buffer boundary and evaluated the total proportion of needleleaf vegetation within the buffer. We compared the needleleaf proportion from the fraction output and the field plot data and calculated the RMSE. We also identified the dominant species on each field plot and compared it with the class with the highest proportion in the fraction images.

To assess the impact of spatial resolution on fraction output, we resampled the AVIRIS-NG image to 10 m and 30 m and assigned each pixel in the fraction which outputs the class value of the dominant vegetation fraction within the pixel. We categorized 40 plots (10 m × 10 m) into three classes, needleleaf, broadleaf and mixed based on the dominant vegetation and then calculated the confusion matrix for fraction outputs (spatial resolutions: 5 m, 10 m, and 30 m). We then compared the user and producer accuracies evaluated using a confusion matrix for all three products as well as for all three classes.

### 2.6.3. Assessment Using High-Resolution (1 m) HySpex Hyperspectral Data

The accuracy assessment of fractional cover is challenging. We devised an approach where we used a ~1000 m$^2$ field plot and a 1 m resolution hyperspectral image (459 bands) acquired using the HySpex hyperspectral camera from a fixed-wing airplane [40]. Figure 7 shows the general methodology for the proposed fraction map validation. We performed random forest classification on the HySpex image using 500 decision trees and 21 features per subset (square root of the total number of bands) [41]. We trained our model using pixels from three classes, needleleaf, broadleaf, and other, and validated the classified map against ~1000 m$^2$ plot data with seven sub-blocks, as shown in Figure 3c. For each block, we recorded the needleleaf tree counts. We calculated the correlation coefficient for the needleleaf count from ground data and the pixels classified as needleleaf on a HySpex classified map. Once we validated the HySpex classified map, we generated 500 random points over the classified map and used the 104 points that fell on the needleleaf class. We then evaluated the needleleaf fraction maps at 5 m, 10 m and 30 m by comparing the points over a needleleaf class with the corresponding pixel in the MESMA outputs.

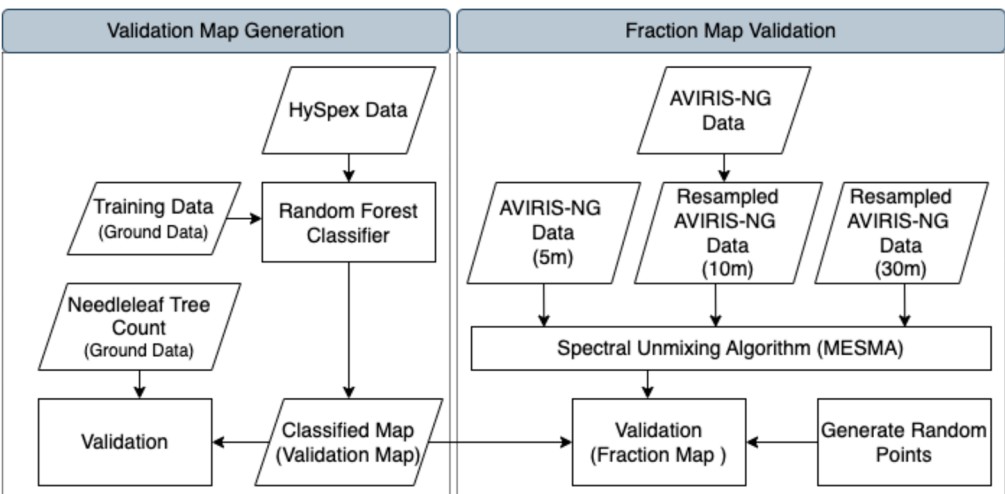

**Figure 7.** Workflow to validate the fraction cover output from AVIRIS-NG data at 5 m, 10 m, and 30 m resolution using 1 m classified map derived from HySpex data.

### 2.6.4. Comparison of Fraction Outputs at Different Spatial Scales

We compared the fraction outputs at different spatial scales (5 m, 10 m, and 30 m) using the McNemar test, a statistical method used to determine whether there is a significant difference between the outputs [42]. The null hypothesis of the McNemar test is that the fraction outputs at different spatial scales are not significantly different. We used Equation (1) to calculate the *z* score and computed the *p*-value using the chi2.cdf function from the scipy.stats Python module [43] at a significance level of $\alpha$ = 0.05. If the *p* value is less than 0.05, we reject the null hypothesis, i.e., there is a significant difference in the fraction output across different spatial scales.

The test statistics (*z* score) for the McNemar test are given by:

$$z = \frac{(|a - b| - 1)}{\sqrt{(a + b)}} \tag{1}$$

where:
- *a*: number of pixels where test 1 (fraction output 1) is positive and test 2 (fraction output 2) is negative;
- *b*: number of pixels where test 1 (fraction output 1) is negative and test 2 (fraction output 2) is positive.

## 3. Results

We successfully generated fractional cover maps at 5 m, 10 m, and 30 m spatial resolution from the AVIRIS-NG image using the MESMA algorithm. Figure 8 shows an RGB image of the fractional cover maps (Red: broadleaf; Green: needleleaf; Blue: NPV), and Figure 9 shows the needleleaf fraction (in white–green shades) for the test site at a 5 m resolution.

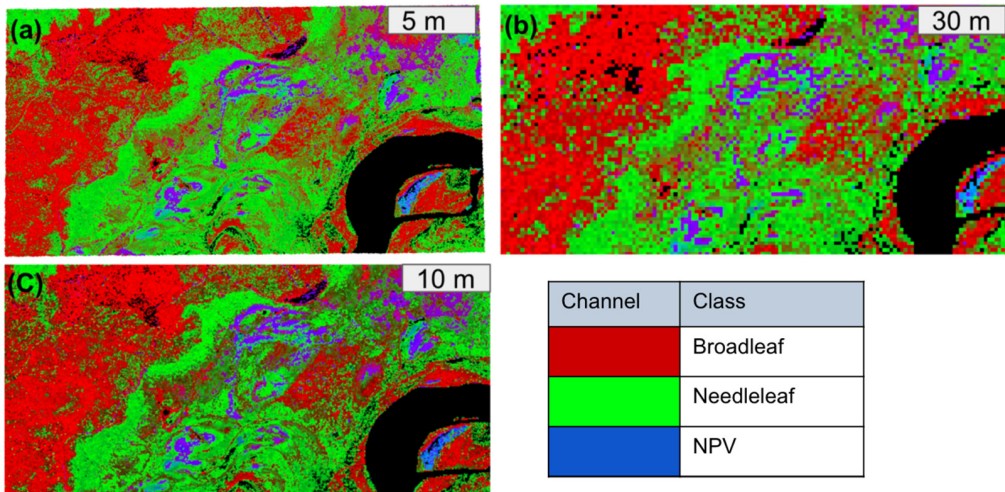

**Figure 8.** RGB images of fraction cover generated using MESMA: (**a**) 5 m spatial resolution; (**b**) 10 m spatial resolution; and (**c**) 30 m spatial resolution.

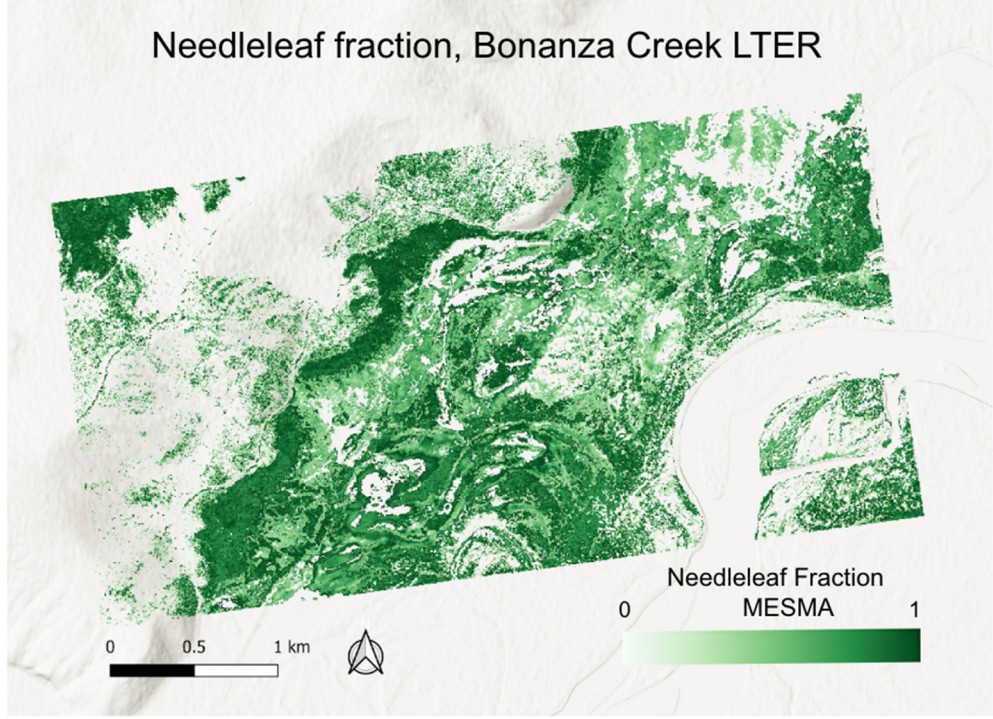

**Figure 9.** Needleleaf fraction cover map at 5 m spatial resolution generated using MESMA on AVIRIS-NG data.

### 3.1. Assessment Using High-Resolution Multispectral Data

We used a very high spatial resolution (50 cm) SkySat image and ground observations to visually compare the fractional map cover outputs and found that the MESMA performed well in capturing the distribution of needleleaf and broadleaf vegetation types. The area dominated by grass is unclassified in the fraction map since we did not use an endmember for grass (Figure 10a). In the color infrared (CIR) band combination, the needleleaf pixels are darker in color, and in the case of the fractional output map, a similar pattern of the needleleaf vegetation is present (in green), highlighted by a white boundary (Figure 10b).

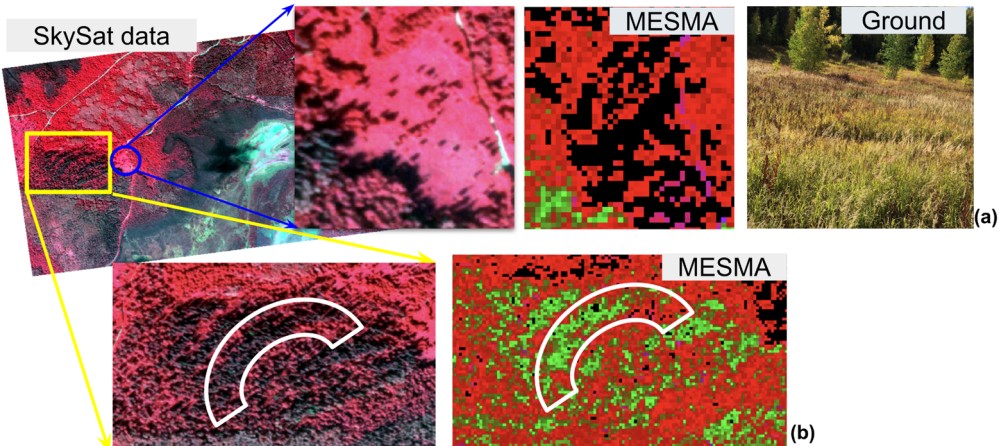

**Figure 10.** Visual comparison of MESMA fraction output using high resolution (50 cm) multispectral data (SkySat data): (**a**) unclassified area where grass was present; and (**b**) similar pattern of needleleaf stands in SkySat image and fraction cover map.

### 3.2. Assessments Using 10 m × 10 m Field Plots

The graph (Figure 11a) shows the comparison between the needleleaf proportion from fraction output and the ground observations (RMSE of 0.34). In the second case, we included a buffer of 10 m to compensate for positional inaccuracy in image data (RMSE reduced to 0.29) (Figure 11b).

Upon comparing the dominant species in each of the 40 plots and the fraction map, we found that 25 plots have the same dominant species as in the fraction map; hence, a total of 62% of plots were correctly mapped. Figure 12 shows the producer and user accuracy for the needleleaf, broadleaf, and other class at different scales using the 40 plots (10 m × 10 m).

### 3.3. Assessment Using High-Resolution (1 m) HySpex Hyperspectral Data

We generated a 1 m high-resolution classified map product using HySpex data to validate the fraction cover. Figure 13a shows the random forest classified map with three classes. Since we are interested in mapping the needleleaf fraction, we performed this validation for needleleaf vegetation. The blue pixels are the needleleaf pixels on the classified map. Figure 13b shows the correlation graph between the number of needleleaf trees based on the ground observation and the number of pixels classified as needleleaf vegetation represented in blue. We found a high positive correlation between them, with a coefficient of 0.9 and an r-squared value of 0.8. We validated the fraction cover map using random points (needleleaf class) and a classified map (validation map). In the case of the 5 m fraction map, 73% of the points were mapped correctly; for the 10 m fraction map, 79% of the points were mapped correctly; and for the 30 m spatial resolution product, 78% of the points were mapped correctly (Figure 14). There was no major difference in accuracy for needleleaf vegetation in all three spatial resolution outputs.

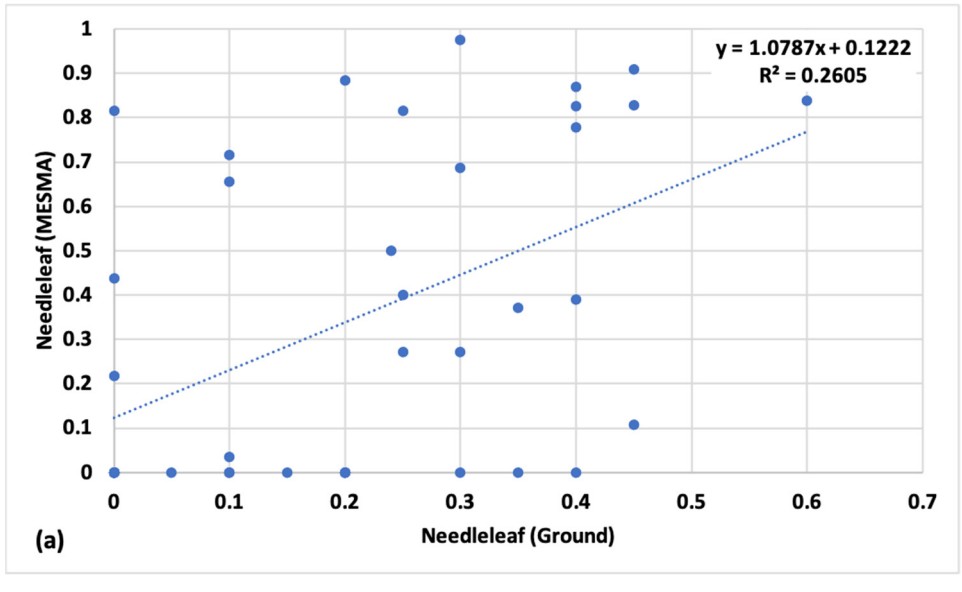

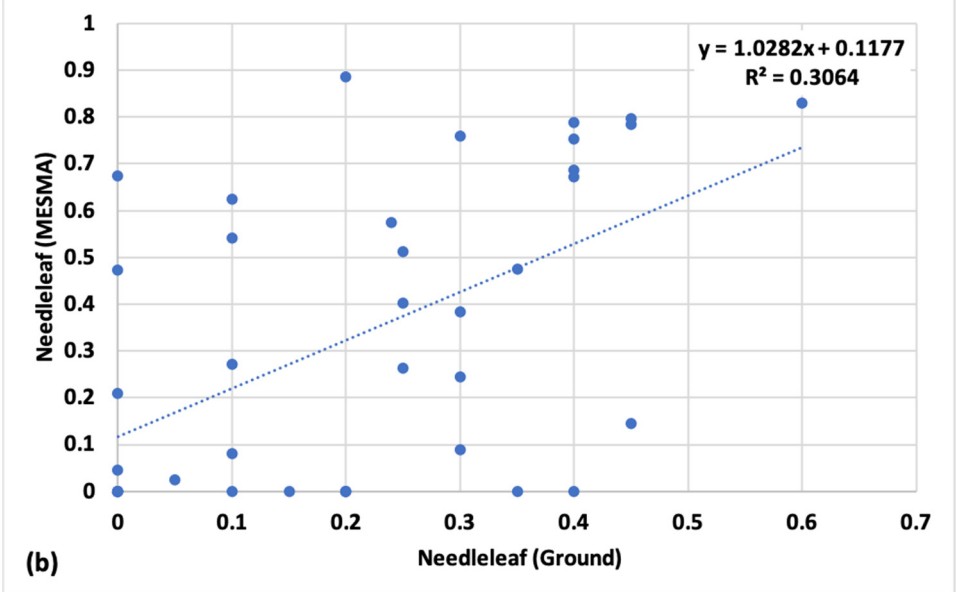

**Figure 11.** Comparison between the needleleaf proportion from the fraction output and the ground data: (**a**) without buffer; and (**b**) with 10 m buffer.

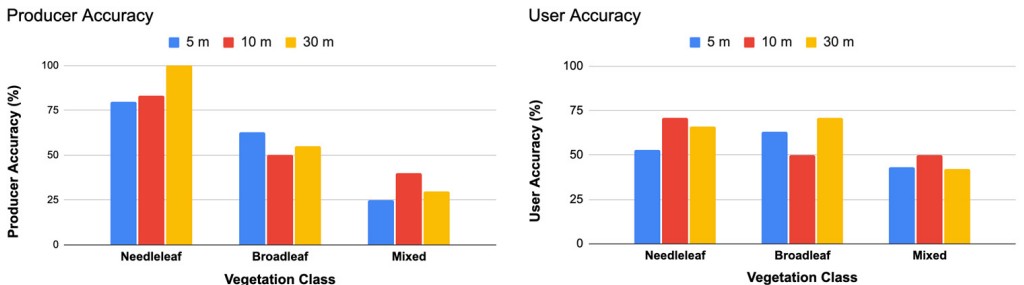

**Figure 12.** Individual class accuracy assessment of fraction cover maps at different spatial scales by assessing the producer accuracy (**left**) and user accuracy (**right**).

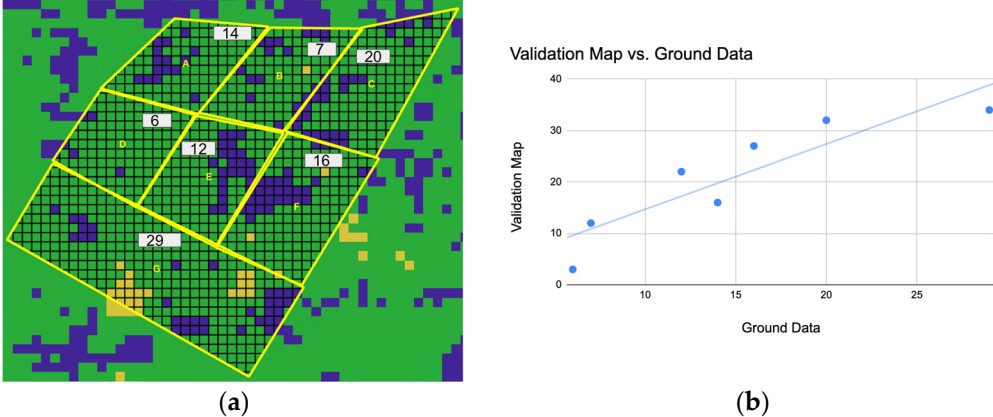

**Figure 13.** (**a**) Random forest classified map (blue: needleleaf; green: broadleaf; yellow: other) of the test site using HySpex Data (1 m spatial resolution). Field plot boundary is shown in yellow, with subplots (A–G); (**b**) shows the correlation between the number of needleleaf trees based on the ground observation and the number of pixels classified as needleleaf class on HySpex Data.

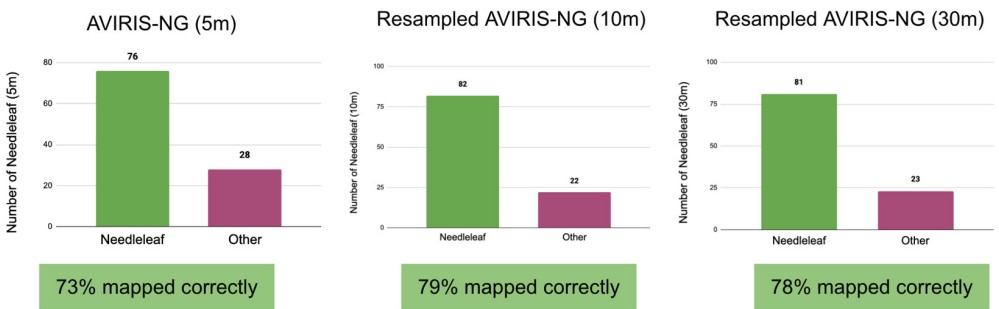

**Figure 14.** Validation of the MESMA fraction cover map (at 5 m, 10 m, and 30 m) using random needleleaf pixels from the HySpex classified map.

### 3.4. Comparison of Fraction Outputs at Different Spatial Scales

We used the McNemar test to determine whether the fraction output results were significantly different from each other. Tables 4–6 show the contingency matrices obtained by applying the McNemar test. Conifer fractions at 5 m and 10 m spatial resolution showed that 72 points were correctly mapped at both scales, resulting in a $z$ score of 1.79 and a $p$ value of 0.18 (Table 4). A comparison of fractions at 5 m and 30 m spatial resolutions, showed that 68 points were mapped correctly at both scales, resulting in a $z$ score of 0.76 and a $p$ value of 0.38 (Table 5). A comparison of fractions at 10 m and 30 m showed an agreement for 70 points, resulting in a $z$ score of 0.00 and a $p$ value of 1.00 (Table 6). In all three cases, the $z$ scores were between $-1.96$ and $1.96$, and the $p$ values were greater than 0.05. These results are consistent with the null hypothesis, indicating that there is no significant difference in the fraction outputs across different spatial scales.

**Table 4.** Contingency matrix for McNemar test to compare fraction outputs (5 m and 10 m).

|  | **5 m Output (Wrong)** | **5 m Output (Correct)** | **All** |
| --- | --- | --- | --- |
| 10 m output (wrong) | 18 | 4 | 22 |
| 10 m output (correct) | 10 | 72 | 82 |
| All | 28 | 76 | 104 |
| McNemar | $z$ score | 1.79 |  |
| results | $p$ value | 0.18 |  |

**Table 5.** Contingency matrix for McNemar test to compare fraction outputs (5 m and 30 m).

|  | 5 m Output (Wrong) | 5 m Output (Correct) | All |
|---|---|---|---|
| 30 m output (wrong) | 15 | 8 | 23 |
| 30 m output (correct) | 13 | 68 | 81 |
| All | 28 | 76 | 104 |
| McNemar | $z$ score | 0.76 | |
| results | $p$ value | 0.38 | |

**Table 6.** Contingency matrix for McNemar test to compare fraction outputs (10 m and 30 m).

|  | 10 m Output (Wrong) | 10 m Output (Correct) | All |
|---|---|---|---|
| 30 m output (wrong) | 11 | 12 | 23 |
| 30 m output (correct) | 11 | 70 | 81 |
| All | 22 | 82 | 104 |
| McNemar | $z$ score | 0.00 | |
| results | $p$ value | 1.00 | |

## 4. Discussion

Most space-borne hyperspectral data are available at a coarser spatial resolution (10 m and 30 m) so there is a need for sub-pixel estimates of the highly flammable fuels to aid in fire management. In this study, we tested the potential of MESMA to map the needleleaf fraction in the boreal region of Alaska and assessed the impact of spatial resolution on the MESMA output. A few studies have applied MESMA for mapping green vegetation, soil, and non-photosynthetic vegetation [18,44,45], but this is the first work in which MESMA has been used to map the needleleaf fraction in a mixed boreal forest of Alaska. The brightness of the needleleaf and broadleaf vegetation is crucial to distinguish between these two classes. In a previous study by Wetherley et al. (2018), the MESMA shade factor was set to 20% to address the variations in brightness between trees and turfgrass in the urban environment of Los Angeles, California [33]. However, we found that limiting the MESMA shade factor to 30% is more effective to separate the needleleaf from broadleaf vegetation.

Validation at the pixel level commonly involves using the pixel value from the centroid of the field plot [19]. However, in mixed and highly diverse boreal settings, this method is not suitable. Therefore, we calculated the proportion of each pixel within a plot and summed up all the proportions for the ground-based validation (Figure 6). Assessing the fraction map accuracy using traditional ground-based methods has several challenges. One of the main challenges was accurately aligning the ground plot with the corresponding pixel, which was particularly difficult in a dense forest area. Furthermore, a precise estimation of the proportion of different vegetation species during the field surveys at the 10–30 m scale was challenging due to the inherent nature of the forested areas. Therefore, we approximated the proportion of the different species on the ground and performed the validation (Figure 11). The approach had limitations and proved ineffective for the needleleaf fraction accuracy assessment. Given the limitations of this validation method, we developed and tested an alternative approach, which is explained in Section 3.3.

For a qualitative assessment, we visually compared the fraction cover output with the 50 cm resolution SkySat image data and observed that the fraction map captured the patterns of needleleaf vegetation distribution reasonably well. Additionally, for quantitative assessment, we collected a ~1000 m² plot, divided it into seven sub-plots (Figure 3c), and counted trees by species in each subplot. A similar approach was used by Fernández-García et al. (2021), where they used the high-resolution aerial orthophoto in place of field plots to establish plots of 30 m × 30 m, and subdivided each plot into 100 cells of 3 m × 3 m [19]. However, collecting similar in situ plot data in a dense forest setting poses a significant challenge. Due to the unavailability of aerial orthophotos, we collected ground data and divided the ~1000 m² plot into seven subplots, counting the trees in each subplot. The trees of the same species were similar in shape and size due

to belonging the same age group. These data were used to validate the high-resolution hyperspectral product (HySpex: 1 m spatial resolution). Assuming that one pixel of HySpex (1 m) corresponds to one tree, we computed Pearson's correlation coefficient which was very high (0.9), implying that the classified output (validation map) is reliable and can be used to validate the fraction output.

We assessed the accuracy of the MESMA output at different spatial resolutions using random points and found no major differences in accuracy at different spatial resolutions. This suggests that MESMA can be used for improved sub-pixel cover estimates from the current and upcoming global space-borne hyperspectral data of coarser resolution. Roth et al. (2015) published a similar study to assess the impact of spatial resolution on the plant functional type classification over five different ecosystems in the USA, including three forest ecosystems: the Smithsonian Environmental Research Center (Maryland), the Wind River Experimental Forest (Washington), and the Sierra National Forest (California), a site with tidal marsh in the Gulf Coast (Louisiana), as well as the central coast region of Santa Barbara, the Santa Ynez Mountains, and the Santa Ynez Valley (California), which feature diverse habitats ranging from evergreen and deciduous shrublands to open grasslands and woodlands. They performed pixel-level classification and suggested that the plant functional type classification will be efficient in the current and upcoming (30 m and 60 m) coarser resolution hyperspectral data [25]. Similarly, we found that there was no significant difference in the fraction outputs at different spatial scales (5 m, 10 m, and 30 m), suggesting that a spectral unmixing technique (MESMA) is effective in estimating the sub-pixel needleleaf fraction from coarser spatial resolution data.

This study presents an effective approach to map and validate the sub-pixel needleleaf fraction in a boreal forest to aid fire management. Our validation approach introduces a novel methodology that will benefit future research on spectral unmixing validation at the sub-pixel level. This intermediate step can serve as a valuable tool for validating very-coarse-resolution orbital data from sensors such as EMIT and SBG.

The fraction map provides detailed information on needleleaf vegetation, which is essential for fire management in all three stages (pre-fire, active fire, and post-fire). It can help land managers and firefighters identify the location of high-risk fuels and employ pre-fire management practices: during an active fire, it will help prioritize the area that needs immediate attention to reduce the fire spread; post-fire, these maps can also help in studying the forest demography changes, especially the forest recovery. While we validated the fraction map in a typical boreal ecosystem, this approach can be used to validate the MESMA output in other ecosystems. The requirement is for detailed field surveys and high-resolution hyperspectral data over the same region.

## 5. Conclusions

Needleleaf vegetation is a high-risk fire fuel and responsible for rapid fire spread and high burn intensity. This study presents an effective approach to quantify the needleleaf fraction in each pixel of an AVIRIS-NG image using a well-documented pixel unmixing algorithm (MESMA) and validating the fraction estimates. We developed an approach to validate a fraction map product using a high-resolution classified map product and needleleaf tree counts from the field. We found that MESMA has the potential to map a needleleaf fraction in a mixed boreal forest with reasonable accuracy. We applied MESMA on AVIRIS-NG data at different spatial resolutions and found no major difference in accuracies suggesting that spectral unmixing is effective in estimating the needleleaf fraction from coarse-resolution data. Future research should focus on the different unmixing techniques and compare their performance. The findings from this study supports the applications of the current and upcoming hyperspectral space missions for sub-pixel vegetation and landcover mapping for a variety of applications including wildfire management and ecosystem monitoring.

**Author Contributions:** Conceptualization, A.B., S.K.P. and D.R.T.; methodology, A.B., S.K.P., D.A.R. and D.R.T.; software, A.B. and D.A.R.; validation, A.B., S.K.P. and D.R.T.; formal analysis, A.B.; investigation, A.B.; resources, S.K.P., D.R.T. and U.S.B.; data curation, A.B., S.K.P., D.R.T. and C.F.W.; writing—original draft preparation, A.B. and S.K.P.; writing—review and editing, A.B., S.K.P., D.A.R., D.R.T., U.S.B. and C.F.W.; visualization, A.B., S.K.P. and D.R.T.; supervision, S.K.P., D.R.T. and D.A.R.; project administration, S.K.P.; funding acquisition, S.K.P. and U.S.B. All authors have read and agreed to the published version of the manuscript.

**Funding:** This material is based upon work supported by the National Science Foundation under the award OIA-1757348, the State of Alaska, and the U.S. Geological Survey under Grant/Cooperative agreement No. G18AP00077.

**Data Availability Statement:** The data used in this study can be found at https://tinyurl.com/ycxnns7t, and the AVIRIS-NG image is available at https://avirisng.jpl.nasa.gov/dataportal/.

**Acknowledgments:** Part of this research was carried out at the Jet Propulsion Laboratory, California Institute of Technology, under contract with the National Aeronautics and Space Administration. We would like to acknowledge the NASA Commercial Smallsat Data Acquisition (CSDA) Program for providing the SkySat data used in this study. We extend our gratitude to Tanya Harrison (Planet Labs) for her assistance in obtaining the SkySat data. We would also like to thank Francisco Ochoa and Philip G. Brodrick from JPL for their guidance in understanding the unmixing algorithm. Lastly, we express our heartfelt appreciation to Chris Smith, Brooke Kubby, Colleen Haan, Malvika Shriwas, Soumitra Sakhalkar, Glen Woodworth, Naomi Hutchens, Michelle Q and Josh Jones for their invaluable help during the field work.

**Conflicts of Interest:** The authors declare no conflict of interest.

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
