# Peer review of "Estimation and Validation of Sub-Pixel Needleleaf Cover Fraction in the Boreal Forest of Alaska to Aid Fire Management"

_remotesensing, doi:10.3390/rs15102484_

Round 1

Reviewer 1 Report

Comments are given on the paper, attached.

Reviewer 2 Report

See attached file.

Reviewer 3 Report

Manuscript ID

remotesensing-2362711

Title

Estimation and Validation of Sub-Pixel Needleleaf Cover Fraction in the Boreal Forest of Alaska to Aid Fire Management

This article identified needleleaf in the sub-pixel scale by Multiple Endmember Spectral Mixture Analysis (MESMA). It is necessary to estimation subpixel needleleaf cover fraction, while I think the data and method in this study have no more novelty. There are already many spectral unmixing methods in current researches. Authors did not compare the advantages of this method relative to other methods. When the study area is small enough and there is sufficient spectral information in the hyperspectral data, the classification results will naturally be good, and the advantages of the method may not be well demonstrated. What we are more concerned about is whether good identification results can be achieved when the study area is enlarged and more easily obtainable remote sensing data is used.
